# QCD Phase Structure and In-Medium Modifications of Meson Masses in Polyakov Linear-Sigma Model with Finite Isospin Asymmetry [†]

Abdel Nasser Tawfik 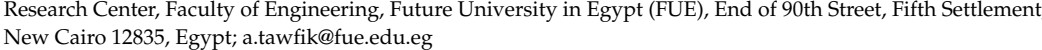

Research Center, Faculty of Engineering, Future University in Egypt (FUE), End of 90th Street, Fifth Settlement, New Cairo 12835, Egypt; a.tawfik@fue.edu.eg

[†] This paper is an extended version from the proceeding paper: Abdel Nasser Tawfik. QCD Phase Structure and In-Medium Modifications of Meson Masses in Polyakov Linear-Sigma Model with Finite Isospin Asymmetry. In Proceedings of the 2nd Electronic Conference on Universe, online, 16 February–2 March 2023; Preprints: ECTP-2023-03, WLCAPP-2023-03, FUE-2023-23.

**Abstract:** In the QCD-like effective model, the Polyakov linear-sigma model, the isospin sigma field ($\bar{\sigma}_3 = f_{K^\pm} - f_{K^0}$) and the third generator of the matrix of the explicit symmetry breaking [$h_3 = m_{a_0}^2 (f_{K^\pm} - f_{K^0})$] are estimated in terms of the decay constants of the neutral ($f_K^0$) and charged Kaon ($f_{K^\pm}$) and the mass of $a_0$ meson. Both quantities $\bar{\sigma}_3$ and $h_3$ are then evaluated, at finite baryon ($\mu_B$), isospin chemical potential ($\mu_I$), and temperature ($T$). Thereby, the dependence of the critical temperature on isospin chemical potential could be mapped out in the ($T - \mu_I$) phase diagram In the QCD-like effective model, the Polyakov linear-sigma model, the isospin sigma field ($\bar{\sigma}_3 = f_{K^\pm} - f_{K^0}$) and the third generator of the matrix of the explicit symmetry breaking [$h_3 = m_{a_0}^2 (f_{K^\pm} - f_{K^0})$] are estimated in terms of the decay constants of the neutral ($f_K^0$) and charged Kaon ($f_{K^\pm}$) and the mass of $a_0$ meson. Both quantities $\bar{\sigma}_3$ and $h_3$ are then evaluated, at finite baryon ($\mu_B$), isospin chemical potential ($\mu_I$), and temperature ($T$). Thereby, the dependence of the critical temperature on isospin chemical potential could be mapped out in the ($T - \mu_I$) phase diagram. The in-medium modifications of pseudoscalars ($J^{pc} = 0^{-+}$), scalars ($J^{pc} = 0^{++}$), vectors ($J^{pc} = 1^{--}$), and axial-vectors ($J^{pc} = 1^{++}$) meson states are then analyzed in thermal and dense medium. We conclude that the QCD phase diagram ($T - \mu_I$) is qualitatively similar to the ($T - \mu_B$) phase diagram. We also conclude that both temperature and isospin chemical potential enhance the in-medium modifications of the meson states $a_0$, $\sigma$, $\eta'$, $\pi$, $f_0$, $\kappa$, $\eta$, $K$, $\rho$, $\omega$, $\kappa^*$, $\phi$, $a_1$, $f_1$, $K^*$, and $f_1^*$. Regarding their chemical potential, at high temperatures the various meson states likely dissolve into colored partonic phase. In this limit, the meson masses form a universal bundle. Thus, we conclude that the increase in the chemical potential similar to temperature derives the colorless confined meson states into the colored deconfined parton phase.

**Keywords:** chiral symmetries; isospin asymmetry; chiral phase structure; in-medium modifications of meson masses

**PACS:** 11.30.Rd; 11.10.Wx; 12.39.Fe; 21.10.Hw





## 1. Introduction

We analyze the genuine finite density and thermal in-medium effects on QCD-like effective theories coupled with the Polyakov-loop potentials for light and strange quark flavors, the quark-hadron phase structure of QCD matter from SU(4) [1,2], thermodynamics and magnetization [3], chiral phase structure and meson masses in finite magnetic field [4], and the in-medium modification of sixteen mesonic states [5]. We introduce the isospin chemical potential to the Polyakov linear-sigma model and present a genuine estimation of the isospin sigma field as well as the third generator of the matrix of the explicit symmetry breaking. Due to the spontaneous symmetry breaking in this QCD-like effective

nonperturbative approach, the mean value of the Polyakov fields $\Phi$, $\langle\Phi\rangle$, and $\langle\Phi^\dagger\rangle$ could be related to the quantum numbers of the vacuum [6]. Thereby, we find that the expectation values of $\bar\pi_a$ vanish, while those of the quark condensates $\bar\sigma_a$ are finite, so that the diagonal generators $U(3)$ correspond to the expectation values $\bar\sigma_0 \neq \bar\sigma_3 \neq \bar\sigma_8 \neq 0$. In this regard, we emphasize that finite $\mu_I$ likely forms a Bose–Einstein Condensate (BEC) phase, especially at the chemical potential aceeding the pion mass, $\mu_c \sim m_\pi$. In the SU(2) formulation, the solution for the minimum of the free energy is achieved for a nonzero value of the pion field, i.e., finite pions condense. In the SU(3) formulation, the same is also likely for kaons. There is abundant evidence and results about the phase of both condensations in the lattice QCD simulations and also in the theoretical models [7–14]. As the present analysis only remains below BEC, we did not calculate BEC, although some of the relevant features concerning finite $mu_I$ would also happen in the BEC phase. Such an analysis shall be carried out elsewhere.

In SU(3), the finite diagonal components of the matrix $H = T_a\,h_a$, where $T_a = \hat\lambda_a/2$ and $a \in \{0, 1, \cdots, 8\}$ are nine U(3) generators, namely, $h_0$, $h_3$, $h_8$, lead to finite quark condensates $\bar\sigma_0$, $\bar\sigma_3$ and $\bar\sigma_8$ corresponding to the three quark flavors. These are the expectation values. Thereby, the masses of the three quark flavors are no longer degenerate, namely, $m_u \neq m_d \neq m_s$. From the orthogonal basis transformation of the original basis, $\sigma_0$, $\sigma_3$, and $\sigma_8$ to pure up ($\sigma_u$), down ($\sigma_d$), and strange ($\sigma_s$) quark flavor basis, the light and strange quark condensates, respectively, could be converted as

$$\begin{bmatrix}\bar\sigma_u\\\sigma_d\\\bar\sigma_s\end{bmatrix} = \frac{1}{\sqrt{3}}\begin{bmatrix}\sqrt{2} & 1 & 1\\\sqrt{2} & -1 & 1\\1 & 0 & -\sqrt{2}\end{bmatrix}\begin{bmatrix}\bar\sigma_0\\\sigma_3\\\bar\sigma_8\end{bmatrix}. \tag{1}$$

In this regard, we emphasize that the strange quark chemical potential is determined under the condition of the vanishing strange quantum number, i.e., equal strange and anti-strange number densities [15–18]. The corresponding masses are given as $m_u = g\sigma_u/2$, $m_d = g\sigma_d/2$, and $m_s = g\sigma_s/\sqrt{2}$, where $g$ is the Yukawa coupling constant. For non-vanishing $\bar\sigma_3$, i.e., $\sigma_u = \sigma_l + \sigma_3$ and $\sigma_d = \sigma_l - \sigma_3$, the effects of finite isospin asymmetry on the $u$- and $d$-quark condensates can be studied. This allows one to estimate the pure mesonic potential of $N_f$ quark flavors, Equation (9), in thermal and dense medium. The derivation of $h_3$, the third generator of the matrix of the explicit symmetry breaking $H = T_a\,h_a$, and the isospin sigma field $\bar\sigma_3$, are introduced in references [19,20]. This is the theoretical framework for the thermal and dense dependence of both quantities, $h_3$ and $\bar\sigma_3$.

At nonzero $\mu_I$, $T$, $\mu_S$, $\mu_B$, a chiral perturbation theory framework with quite compelling results compared to lattice simulations has been reported [7–14]. The Polyakov linear-sigma model (PLSM) as a QCD-like approach is well developed [21–30]. The author contributed extensive studies of the higher-order moments [31,32], the QCD matter at finite magnetic field [33–35], the chiral QCD phase transition [36], the bulk and shear viscosity properties [37], the conductivity properties [38], the chiral magnetic properties [39,40], the chiral phase structure of the sixteen meson states [41], and a comparison between mean-field approximation and optimized perturbation theory [42]. The present script aims at analyzing

- How to properly include finite isospin asymmetry in the QCD-like effective model, PLSM;
- The thermodynamics and QCD phase structure in PLSM; and
- The in-medium modifications of the pseudoscalars ($J^{pc} = 0^{-+}$), scalars ($J^{pc} = 0^{++}$), vectors ($J^{pc} = 1^{--}$), and axial-vectors ($J^{pc} = 1^{++}$) meson states.

The present script is organized as follows. The formalism is introduced in Section 2. In Section 2.1, we first review the QCD-like effective model, as well as the SU(3) Polyakov linear-sigma model (PLSM), at vanishing $h_3$. The generalization to finite isospin asymmetry is discussed in Section 2.2. In Section 2.3, the in-medium modifications of pseudoscalars, scalars, vectors, and axial-vectors meson states are derived. In Section 3, the results are

discussed. The results of thermodynamics and chiral phase transition are introduced in Section 3.1. In Section 3.2, the in-medium modifications of $a_0$, $\sigma$, $\eta'$, $\pi$, $f_0$, $\kappa$, $\eta$, $K$, $\rho$, $\omega$, $\kappa^*$, $\phi$, $a_1$, $f_1$, $K^*$, and $f_1^*$ meson states in thermal and dense medium are presented. Section 4 is devoted to the conclusions.

## 2. Formalism

### 2.1. PLSM with Finite Chemical Potentials and Vanishing Isospin Asymmetry

The Lagrangian of the linear-sigma model (LSM) with $N_f$ quark flavors and the Polyakov-loop potential can be summarized as $\mathcal{L}_{PLSM} = \mathcal{L}_\psi + \mathcal{L}_m - \mathcal{U}(\phi, \bar{\phi}, T)$, where $\mathcal{L}_\psi$ represents the quark (fermion) part, $\mathcal{L}_m$ stands for the meson (boson) part, and $\mathcal{U}(\phi, \bar{\phi}, T)$ is the Polyakov-loop contribution to the PLSM potential.

The contributions of the quarks (fermions) to the PLSM potential can be expressed as

$$\mathcal{L}_\psi = \sum_f \overline{\psi}_f (i\gamma^\mu D_\mu - g\, T_a(\sigma_a + i\gamma_5 \pi_a))\psi_f, \tag{2}$$

where $\psi$ are the Dirac spinor fields; $f = [u, d, s]$ are the quark flavors; and $D_\mu$, $\mu$, $\gamma^\mu$ represent the covariant derivative, Lorentz index, and chiral spinors. The contributions of the mesons (bosons) to the PLSM potential are given as

$$\begin{aligned}\mathcal{L}_m &= \mathrm{Tr}(\partial_\nu \Phi^\dagger \partial^\nu \Phi - m^2 \Phi^\dagger \Phi) - \lambda_1 [\mathrm{Tr}\,(\Phi^\dagger \Phi)]^2 - \lambda_2 \,\mathrm{Tr}(\Phi^\dagger \Phi)^2 \\ &+ c[\mathrm{Det}(\Phi) + \mathrm{Det}(\Phi^\dagger)] + \mathrm{Tr}[H(\Phi + \Phi^\dagger)],\end{aligned} \tag{3}$$

where $\Phi$ is the nonet meson $(3 \times 3)$-matrix,

$$\Phi = \sum_{a=0}^{N_f^2-1} T_a(\bar{\sigma}_a + i\bar{\pi}_a). \tag{4}$$

and $T_a = \hat{\lambda}_a/2$ is a generator operator in U(3) algebra. $T_a$ can be determined from the Gell–Mann matrices $\hat{\lambda}_a$ [43].

Before we outline the third type of contribution to the PLSM potential, a few remarks are in order. In LSM, there are different parameters to be fixed, $m^2$, $h_l$, $h_s$, $\lambda_1$, $\lambda_2$, and $c$, depending on the sigma meson mass $m_\sigma$ [44]. Table 1 summarizes the values of these parameters used in the present calculations, at $m_\sigma = 800$ MeV [44]. The definitions of $h_a$ are given in Equations (21)–(23).

**Table 1.** The LSM parameters which are fixed at $m_\sigma = 800$ MeV and $h_3 = 0$ [44].

| $m_\sigma$ [MeV] | $c$ [MeV] | $h_{ud}$ [MeV$^3$] | $h_3$ [MeV$^3$] | $h_s$ [MeV$^3$] | $m^2$ [MeV$^2$] | $\lambda_1$ | $\lambda_2$ |
|---|---|---|---|---|---|---|---|
| 800 | 4807.84 | $(120.73)^3$ | 0 | $(336.41)^3$ | $-(306.26)^2$ | 13.49 | 46.48 |

The third type of contribution to the PLSM Lagrangian is from the Polyakov-loop potentials, the gluonic degrees of freedom responsible for the dynamics of the quark-gluon interactions, $\mathcal{U}(\phi, \bar{\phi}, T)$. There is no pure theoretical prediction of the Polyakov-loop potentials. These potentials are rather suggested from the QCD symmetries in pure-gauge theory [22,23,28,45,46]. From the strong coupling simulations including the higher-order Polyakov-loop variables,

$$\mathcal{U}_{\mathrm{Fuku}}(\phi, \bar{\phi}, T) = -b\, T\left[54\, \phi\, \bar{\phi}\, \exp(-a/T) + \ln(1 - 6\phi\bar{\phi} - 3(\phi\bar{\phi})^2 + 4(\phi^3 + \bar{\phi}^3))\right], \tag{5}$$

has been suggested in Ref. [22]. The thermal expectation value of the color traced Wilson loop, the Polyakov-loop variable, is then given as

$$\phi = (\mathrm{Tr}_c\, \mathcal{P})/N_c, \qquad \bar{\phi} = (\mathrm{Tr}_c\, \mathcal{P}^\dagger)/N_c, \tag{6}$$

where $\mathcal{P}$ represents the Polyakov loops.

Now, we formulate the PLSM grand-canonical potential in either mean-field approximation [31] or optimized perturbation theory [42]. From here onward, the expectation values of the $\sigma$-fields shall be written unbarred.

$$\Omega(T, \mu_f) = \frac{-T\,\cdot\ln\left[\mathcal{Z}\right]}{V} \;\; = \;\; \Omega_{\bar{\psi}\psi}(T,\,\mu_f) + U(\sigma_u, \sigma_d, \sigma_s) + \mathcal{U}_{\mathrm{Fuku}}(\phi, \bar{\phi}, T), \tag{7}$$

where $\mu_f$ is the chemical potentials of the three quark flavors. $\mu_f$ counts for all types of chemical potentials and therefore plays a crucial role in the present study. It should be emphasized that $U(\sigma_u, \sigma_d, \sigma_s)$ also depends on both $T$ and $\mu_f$ through the quark condensations. For the sake on completeness, it is worth mentioning that the analytic form of the partition function, Equation (7), is indeed justified. Since the result resembles the free fermion gas, at least, without the Polyakov corrections, the perturbation theory in the QCD coupling constant has been assumed.

For the conserved quantum numbers, baryon $B$, strangeness $S$, electric charge $Q$, and isospin $I$, the quark chemical potentials are constructed as

$$\mu_u = \frac{\mu_B}{3} + \frac{2\mu_Q}{3} + \frac{\mu_I}{2}, \qquad \mu_d = \frac{\mu_B}{3} - \frac{\mu_Q}{3} - \frac{\mu_I}{2}, \qquad \mu_s = \frac{\mu_B}{3} - \frac{\mu_Q}{3} - \mu_S. \tag{8}$$

This allows one to define the mesonic contributions to the LSM potential $U(\sigma_u, \sigma_d, \sigma_s)$. By substituting Equation (4) into Equation (3), we obtain

$$\begin{aligned} U(\sigma_u, \sigma_d, \sigma_s) \;\; = \;\; & \frac{m^2}{4}\left[\sigma_u^2 + \sigma_d^2 + 2\sigma_s^2\right] - \frac{c}{2\sqrt{2}}\sigma_u\sigma_d\sigma_s + \frac{\lambda_1}{16}\left(\sigma_u^2 + \sigma_d^2 + 2\sigma_s^2\right)^2 \\ + \;\; & \frac{\lambda_2}{16}\left(\sigma_u^4 + \sigma_d^4 + 4\sigma_s^4\right) - h_{ud}\frac{\sigma_u + \sigma_d}{2} - h_s\sigma_s - h_3\frac{\sigma_u - \sigma_d}{2}. \end{aligned} \tag{9}$$

The second line expresses the $h_3$ term contributing to the isospin asymmetry. An estimation of $h_3$ shall be derived in Section 2.2.

The quark and antiquark contributions are then given as [22,47–49]

$$\Omega_{\bar{\psi}\psi}(T, \mu_f) \;\; = \;\; -2\,T \sum_{f=u,d,s} \int_0^\infty \frac{d^3\vec{P}}{(2\pi)^3} \ln\left[1 + n_{q,f}(T,\,\mu_f)\right] + \ln\left[1 + n_{\bar{q},f}(T,\,\mu_f)\right]. \tag{10}$$

where the number density is given as

$$n_{q,f}(T, \mu_f) \;\; = \;\; 3\left(\phi + \bar{\phi}e^{-\frac{E_f - \mu_f}{T}}\right) \times e^{-\frac{E_f - \mu_f}{T}} + e^{-3\frac{E_f - \mu_f}{T}}, \tag{11}$$

where $E_f = (\vec{P}^2 + m_f^2)^{1/2}$, the energy-momentum dispersion relation corresponds to the quark and antiquark, and $m_f$ is the mass of $f$-th quark flavor . Both $n_{\bar{q},f}(T, \mu_f)$ and $n_{q,f}(T, \mu_f)$ are identical if $-\mu_f$ is replaced by $+\mu_f$ and $\phi$, the order parameter of the Ployakov-loop field, by its conjugate $\bar{\phi}$ or vice versa .

### 2.2. PLSM with Finite Chemical Potentials and Isospin Asymmetry

In SU(2), the isospin asymmetry is broken at non-vanishing $\bar{\sigma}_3$ [6]. Moreover $\bar{\sigma}_3$, the potential of mesonic contributions in SU($N_f$) [50],

$$U(\bar{\sigma}) \;\; = \;\; \left(\frac{m^2}{2} - h_a\right)\bar{\sigma}_a - 3\mathcal{G}_{abc}\bar{\sigma}_b\,\bar{\sigma}_c - \frac{4}{3}\mathcal{F}_{abcd}\,\bar{\sigma}_b\,\bar{\sigma}_c\bar{\sigma}_d, \tag{12}$$

also breaks the isospin asymmetry. Both coefficients $\mathcal{G}_{abc}$ and $\mathcal{F}_{abcd}$ are given as [50]

$$\mathcal{G}_{abc} = \frac{c}{6}\left[d_{abc} - \frac{3}{2}(d_{0bc}\delta_{a0} + d_{a0c}\delta_{b0} + d_{ab0}\delta_{c0}) + \frac{9}{2}d_{000}\delta_{a0}\delta_{b0}\delta_{c0}\right], \tag{13}$$

$$\mathcal{F}_{abcd} = \frac{\lambda_1}{4}[\delta_{ab}\delta_{cd} + \delta_{ad}\delta_{cd} + \delta_{ac}\delta_{bd}] + \frac{\lambda_2}{8}[d_{abn}d_{ncd} + d_{adn}d_{nbc} + d_{acn}d_{nbd}]. \tag{14}$$

We remark that the meson potential is only included at tree level. On the other hand, the perturbation in the $\lambda_1$ couplings is meant to give rise to large contributions. When limiting the discussion to the non-perturbative regime, i.e., the regime covered by the non-perturbative lattice QCD simulations, the consistency of this framework is guaranteed.

The symmetry breaking terms, $h_0$, $h_3$, and $h_8$, are to be determined from the minimization of the PLSM potential, Equation (12). On a tree level, we assume $\partial U(\bar{\sigma})/\partial\bar{\sigma}_a = 0$. We remark that $h_0$ and $h_8$ could be determined from the partially conserved axial current (PCAC) relations. For $h_3$, we recall that the generator operator $\hat{T}_a = \hat{\lambda}_a/2$ in U(3) can be obtained from the Gell–Mann matrices $\hat{\lambda}_a$ [43],

$$[\hat{T}_a, \hat{T}_b] = if_{abc}\hat{T}_c, \qquad \{\hat{T}_a, \hat{T}_b\} = id_{abc}\hat{T}_c, \tag{15}$$

where $f_{abc}$ and $d_{abc}$ are the standard antisymmetric and symmetric structure constants of SU(3),

$$d_{abc} = \frac{1}{4}Tr[\{\hat{\lambda}_a, \hat{\lambda}_b\}\hat{\lambda}_c], \qquad d_{ab0} = \sqrt{\frac{2}{3}}\,\delta_{ab}. \tag{16}$$

In the PCAC relation, the decay constant $f_a$ is related to the symmetric structure constant $f_a = d_{aab}\bar{\sigma}_a$.

The decay constants of charged and neutral pion mesons ($f_{\pi^\pm} = f_1$, $f_{\pi^0} = f_3$) and of Kaon mesons ($f_{K^\pm} = f_4$, $f_{K^0} = f_6$) can be summarized as

$$f_{\pi^0} = f_{\pi^\pm} = \sqrt{\frac{2}{3}}\bar{\sigma}_0 + \frac{1}{\sqrt{3}}\bar{\sigma}_8, \tag{17}$$

$$f_{K^\pm} = \sqrt{\frac{2}{3}}\bar{\sigma}_0 + \frac{1}{2}\bar{\sigma}_3 - \frac{1}{2\sqrt{3}}\bar{\sigma}_8, \tag{18}$$

$$f_{K^0} = \sqrt{\frac{2}{3}}\bar{\sigma}_0 - \frac{1}{2}\bar{\sigma}_3 - \frac{1}{2\sqrt{3}}\bar{\sigma}_8. \tag{19}$$

Then, we algebraically deduce the isospin sigma field, $\bar{\sigma}_3$. This is obtained as the difference between the decay constants of neutral and charged Kaon mesons,

$$\bar{\sigma}_3 = f_{K^\pm} - f_{K^0}. \tag{20}$$

From the recent experimental and lattice simulations of the physical constants [51–53], we find that $f_{\pi^\pm} = f_{\pi^0} = 92.4$ MeV and $f_{K^\pm} = 113$ MeV, $f_{K^0} = 113.453$ MeV. Then, for $h_0$ and $h_8$, the following expressions are suggested,

$$h_0 = \frac{1}{\sqrt{6}}\left(m_\pi^2 f_\pi + 2m_K^2 f_K\right), \qquad h_8 = \frac{2}{\sqrt{3}}\left(m_\pi^2 f_\pi - m_K^2 f_K\right). \tag{21}$$

The third generator of the matrix of the explicit symmetry breaking of the matrix $H = T_a h_a$, as well as the explicit symmetry breaking term, $h_3$, can then be derived from $\partial U(\bar{\sigma})/\partial\bar{\sigma}_3 = 0$,

$$\begin{aligned}
h_3 = & \left[m^2 + \frac{c}{\sqrt{6}}\bar{\sigma}_0 - \frac{c}{\sqrt{3}}\bar{\sigma}_8 + \lambda_1\left(\bar{\sigma}_0{}^2 + \bar{\sigma}_3{}^2 + \bar{\sigma}_8{}^2\right)\right. \\
& \left. + \lambda_2\left(\bar{\sigma}_0{}^2 + \frac{\bar{\sigma}_3{}^2}{2} + \frac{\bar{\sigma}_8{}^2}{2} + \sqrt{2}\bar{\sigma}_0\bar{\sigma}_8\right)\right]\bar{\sigma}_3,
\end{aligned} \tag{22}$$

where the square brackets $[\cdots]$ are nothing but the squared mass of the $a_0$ meson [19,20]. Then, with Equation (20)

$$h_3 \;=\; m_{a_0}^2 (f_{K^\pm} - f_{K^0}) \tag{23}$$

*2.3. PLSM: In-Medium Modifications of Pseudoscalars, Scalars, Vectors, and Axial-Vectors Meson States*

The meson (hadron) states that could be generated in PLSM are related to the degrees of freedom integrated in. Accordingly, Equation (2) could be generalized as [5],

$$\mathcal{L}_f = \bar{q}\Big[i\slashed{\partial} - g\, T_a \left(\sigma_a + i\,\gamma_5\,\pi_a + \gamma_\zeta V_a^\zeta + \gamma_\zeta \gamma_5 A_a^\zeta\right)\Big]q. \tag{24}$$

Additionally, Equation (3) could be extended to count for the various nonet states, the interactions, and the possible anomalies,

$$\mathcal{L}_m \;=\; \mathcal{L}_{SP} + \mathcal{L}_{VA} + \mathcal{L}_{Int} + \mathcal{L}_{U(1)_A}, \tag{25}$$

where $\mathcal{L}_{SP}$ stands for scalars ($J^{pc} = 0^{++}$) and pseudoscalars ($J^{pc} = 0^{-+}$), while $\mathcal{L}_{AV}$ represents vectors ($J^{pc} = 1^{--}$) and axial-vectors ($J^{pc} = 1^{++}$) meson states. $\mathcal{L}_{Int}$ represents the interactions, from which an anomalous term emerges, $\mathcal{L}_{U(1)_A}$,

$$
\begin{aligned}
\mathcal{L}_{SP} \;=\;& \mathrm{Tr}\Big[(D^\mu \Phi)^\dagger (D^\mu \Phi) - m^2 \Phi^\dagger \Phi\Big] - \lambda_1 [\mathrm{Tr}(\Phi^\dagger \Phi)]^2 - \lambda_2 \mathrm{Tr}(\Phi^\dagger \Phi)^2 \\
&+\; \mathrm{Tr}[H(\Phi + \Phi^\dagger)],
\end{aligned}
\tag{26}
$$

$$
\begin{aligned}
\mathcal{L}_{AV} \;=\;& -\frac{1}{4}\mathrm{Tr}(L_{\mu\nu}^2 + R_{\mu\nu}^2) + \mathrm{Tr}\left[\left(\frac{m_1^2}{2} + \Delta\right)(L_\mu^2 + R_\mu^2)\right] \\
&+\; i\frac{g_2}{2}\left(\mathrm{Tr}\{L_{\mu\nu}[L^\mu, L^\nu]\} + \mathrm{Tr}\{R_{\mu\nu}[R^\mu, R^\nu]\}\right) \\
&+\; g_3[\mathrm{Tr}(L_\mu L_\nu L^\mu L^\nu) + \mathrm{Tr}(R_\mu R_\nu R^\mu R^\nu)] + g_4[\mathrm{Tr}(L_\mu L^\mu L_\nu L^\nu) + \mathrm{Tr}(R_\mu R^\mu R_\nu R^\nu)] \\
&+\; g_5\,\mathrm{Tr}(L_\mu L^\mu)\,\mathrm{Tr}(R_\nu R^\nu) + g_6[\mathrm{Tr}(L_\mu L^\mu)\,\mathrm{Tr}(L_\nu L^\nu) + \mathrm{Tr}(R_\mu R^\mu)\,\mathrm{Tr}(R_\nu R^\nu)],
\end{aligned}
\tag{27}
$$

$$
\mathcal{L}_{Int} \;=\; \frac{h_1}{2}\mathrm{Tr}(\Phi^\dagger \Phi)\,\mathrm{Tr}(L_\mu^2 + R_\mu^2) + h_2\,\mathrm{Tr}[|L_\mu \Phi|^2 + |\Phi R_\mu|^2] + 2h_3\,\mathrm{Tr}(L_\mu \Phi R^\mu \Phi^\dagger), \tag{28}
$$

$$
\begin{aligned}
\mathcal{L}_{U(1)_A} \;=\;& c[\mathrm{Det}(\Phi) + \mathrm{Det}(\Phi^\dagger)] + c_0[\mathrm{Det}(\Phi) - \mathrm{Det}(\Phi^\dagger)]^2 + c_1[\mathrm{Det}(\Phi) \\
&+\; \mathrm{Det}(\Phi^\dagger)]\,\mathrm{Tr}[\Phi\Phi^\dagger].
\end{aligned}
\tag{29}
$$

The scalars $\sigma_a$, i.e., $J^{PC} = 0^{++}$; pseudoscalars $\pi_a$, i.e., $J^{PC} = 0^{-+}$; vectors $V_a^\mu$, i.e., $J^{PC} = 1^{--}$; and axial-vectors $A_a^\mu$, i.e., $J^{PC} = 1^{++}$ can be deduced from

$$\Phi = \sum_{a=0}^{N_f^2 - 1} T_a(\sigma_a + i\pi_a), \qquad L^\mu = \sum_{a=0}^{N_f^2 - 1} T_a\left(V_a^\mu + A_a^\mu\right), \qquad R^\mu = \sum_{a=0}^{N_f^2 - 1} T_a\left(V_a^\mu - A_a^\mu\right). \tag{30}$$

For chiral symmetry breaking in $U(3)_L \times U(3)_R = U(3)_V \times U(3)_A$

$$H = \sum_{a=0}^{8} T_a h_a, \qquad \Delta = \sum_{a=0}^{8} T_a \delta_a. \tag{31}$$

The explicit symmetry breaking stems from

- The finite quark masses in the (pseudo)-scalar and (axial)-vector sectors,
- Breaking $U(3)_A$ if $H_0, \Delta_0 \neq 0$, and
- Breaking $U(3)_V \to SU(2)_V \times U(1)_V$ if $H_8, \Delta_8 \neq 0$.

The covariant derivative, $D^\mu \Phi = \partial^\mu \Phi - i\,g_1(L^\mu \Phi - \Phi R^\mu)$, can be associated with the degrees of freedom of (pseudo-)scalar and (axial-)vector. For simplicity, the isospin asymmetry is neglected. Then, the special choice of $h_a$ as $h_0 \neq 0$, $h_3 = 0$, and $h_8 \neq 0$, as well as of $\delta_a$ as $\delta_0 \neq 0$, $\delta_3 = 0$, and $\delta_8 \neq 0$, leads to

$$T_a \sigma_a = \begin{pmatrix} \frac{1}{\sqrt{2}} a_0^0 + \frac{1}{\sqrt{6}} \sigma_8 + \frac{1}{\sqrt{3}} \sigma_0 & a_0^- & \kappa^- \\ a_0^+ & -\frac{1}{\sqrt{2}} a_0^0 + \frac{1}{\sqrt{6}} \sigma_8 + \frac{1}{\sqrt{3}} \sigma_0 & \bar{\kappa}^0 \\ \kappa^+ & \kappa^0 & -\sqrt{\frac{2}{3}} \sigma_8 + \frac{1}{\sqrt{3}} \sigma_0 \end{pmatrix}, \qquad (32)$$

$$T_a \pi_a = \begin{pmatrix} \frac{1}{\sqrt{2}} \pi^0 + \frac{1}{\sqrt{6}} \pi_8 + \frac{1}{\sqrt{3}} \pi_0 & \pi^- & K^- \\ \pi^+ & -\frac{1}{\sqrt{2}} \pi^0 + \frac{1}{\sqrt{6}} \pi_8 + \frac{1}{\sqrt{3}} \pi_0 & \bar{K}^0 \\ K^+ & K^0 & -\sqrt{\frac{2}{3}} \pi_8 + \frac{1}{\sqrt{3}} \pi_0 \end{pmatrix}, \qquad (33)$$

$$T_a V_a^\mu = \begin{pmatrix} \frac{\omega_0 + \rho^0}{\sqrt{2}} & \rho^+ & K^{\star +} \\ \rho^- & \frac{\omega_0 - \rho^0}{\sqrt{2}} & K^{\star 0} \\ K^{\star -} & \bar{K}^{\star 0} & \omega_8 \end{pmatrix}^\mu, \qquad (34)$$

$$T_a A_a^\mu = \begin{pmatrix} \frac{f_{1_0} + a_1^0}{\sqrt{2}} & a_1^+ & K_1^+ \\ a_1^- & \frac{f_{1_0} - a_1^0}{\sqrt{2}} & K_1^0 \\ K_1^- & \bar{K}_1^0 & f_{1_8} \end{pmatrix}^\mu. \qquad (35)$$

The right hand side of each of these expressions is to multiplied by $\frac{1}{\sqrt{2}}$. As we are limiting the current calculations to the non-perturbative regime, the perturbative contributions to the thermal dependence of meson masses can be discussed elsewhere.

The second derivative of the equation of motion with respect to a specific hadron field determines the mass of that hadron state. We assume that the equation of motion is encoded in the generalized free energy $\Omega(T, \mu_f, \zeta) = -T \ln \mathcal{Z}/V$, which apparently encodes details about that hadron state, whose meson field is represented by $\zeta$. $\Omega(T, \mu_f, \zeta)$ is the generalization of Equation (7). For vanishing quark-antiquark potential contributions to the vacuum Lagrangian, the meson potential, from which the mass matrix is deduced, is expressed as

$$m_{i,ab} = \frac{\partial^2 \Omega(T, \mu_f, \zeta)}{\partial \zeta_{i,a} \partial \zeta_{i,b}}. \qquad (36)$$

where $i$ represents the (pseudo)scalar and (axial)vector meson states and $a, b \in \{0, 1, \cdots, 8\}$. Equation (36) counts for:

- The vacuum contributions, i.e., the meson masses are related to the strange $\sigma_s$ and nonstrange $\sigma_l$ sigma fields (the finite isospin asymmetry $\sigma_l$ is then replaced by the distinguishable $\sigma_u$ and $\sigma_d$) and
- The in-medium modifications of the meson masses are given as

$$\begin{aligned} m_{i,ab}^2 = & \; \nu_c \sum_{f=l,s} \int \frac{d^3 p}{(2\pi)^3} \frac{1}{2E_{q,f}} \left[ \left( n_{q,f} + n_{\bar{q},f} \right) \left( m_{f,ab}^2 - \frac{m_{f,a}^2 m_{f,b}^2}{2E_{q,f}^2} \right) \right. \\ & \left. - \left( b_{q,f} + b_{\bar{q},f} \right) \left( \frac{m_{f,a}^2 m_{f,b}^2}{2E_{q,f}T} \right) \right], \end{aligned} \qquad (37)$$

from which the quark mass derivative with respect to the meson fields, $m_{f,a}^2 \equiv \partial m_f^2 / \partial \zeta_{i,a}$, and the derivative with respect to the meson fields, $m_{f,ab}^2 \equiv \partial m_f^2 / \partial \zeta_{i,a} \partial \zeta_{i,b}$, are listed in Table 1 in Ref. [5]. Then, the antiquark function $b_{\bar{q},f}(T, \mu_f) = b_{q,f}(T, -\mu_f)$, and $b_{q,f}(T, \mu_f) = n_{q,f}(T, \mu_f)(1 - n_{q,f}(T, \mu_f))$. The parameter $\nu_c = 2Nc$ counts for the color degrees of freedom. The in-medium meson mass modifications can be determined from the quark-antiquark contributions to the potential, Equation (10), and the diagonalization of the resulting quark mass matrix [54],

$$m_{i,ab}^2 = \nu_c \sum_{f=l,s} \int \frac{d^3p}{(2\pi)^3} \frac{1}{2E_{q,f}} \left[ (N_{q,f} + N_{\bar{q},f}) \left( m_{f,ab}^2 - \frac{m_{f,a}^2 m_{f,b}^2}{2E_{q,f}^2} \right) \right.$$
$$\left. + (B_{q,f} + B_{\bar{q},f}) \left( \frac{m_{f,a}^2 m_{f,b}^2}{2E_{q,f}T} \right) \right], \tag{38}$$

where

$$N_{q,f} = \frac{\Phi e^{-E_{q,f}/T} + 2\Phi^* e^{-2E_{q,f}/T} + e^{-3E_{q,f}/T}}{1 + 3(\phi + \phi^* e^{-E_{q,f}/T}) e^{-E_{q,f}/T} + e^{-3E_{\bar{q},f}/T}}, \tag{39}$$

$$N_{\bar{q},f} = \frac{\Phi^* e^{-E_{\bar{q},f}/T} + 2\Phi e^{-2E_{\bar{q},f}/T} + e^{-3E_{\bar{q},f}/T}}{1 + 3(\phi^* + \phi e^{-E_{\bar{q},f}/T}) e^{-E_{\bar{q},f}/T} + e^{-3E_{\bar{q},f}/T}}, \tag{40}$$

$$C_{q,f} = \frac{\Phi e^{-E_{q,f}/T} + 4\Phi^* e^{-2E_{q,f}/T} + 3e^{-3E_{q,f}/T}}{1 + 3(\phi + \phi^* e^{-E_{q,f}/T}) e^{-E_{q,f}/T} + e^{-3E_{\bar{q},f}/T}}, \tag{41}$$

$$C_{\bar{q},f} = \frac{\Phi^* e^{-E_{\bar{q},f}/T} + 4\Phi e^{-2E_{\bar{q},f}/T} + 3e^{-3E_{\bar{q},f}/T}}{1 + 3(\phi^* + \phi e^{-E_{\bar{q},f}/T}) e^{-E_{\bar{q},f}/T} + e^{-3E_{\bar{q},f}/T}}. \tag{42}$$

As defined in Ref. [54], $B_{q,f} = 3(N_{q,f})^2 - C_{q,f}$ represents quarks and $B_{\bar{q},f} = 3(N_{\bar{q},f})^2 - C_{\bar{q},f}$ represents antiquarks.

## 3. Results

### 3.1. PLSM: Thermodynamics and QCD Phase Structure at Finite Isospin Asymmetry

Equation (7) expresses the grand canonical potential of PLSM in either mean-field approximation [31] or optimized perturbation theory [42], from which the various physical quantities can be derived. Thereby, the QCD thermodynamics and phase structure can be analyzed in thermal and dense medium. To this end, the thermal and dense dependence of the chiral quark condensations is conjectured to play an essential role. The left-hand panel of Figure 1 depicts the thermal dependence of the normalized chiral quark condensates. The dependence of the normalized chiral quark condensates on the isospon chemical potential, $\mu_I$, is presented in the right-hand panel. The legends are applied to all panels. At vanishing $\mu_I$, both $\sigma_u/\sigma_u^0$ and $\sigma_d/\sigma_d^0$ are apparently identical (solid curve). This is no longer the case at the finite $\mu_I$ middle and bottom panels, where the consequences of the finite isospin parameters, $\sigma_3$ and $h_3$, are switched on. With increasing $\mu_I$, the difference between both curves likely increases, especially as the temperature approaches the region of the phase transition. $\sigma_s/\sigma_s^0$ is not affected by $\mu_I$.

The dependence of normalized chiral quark condensates on the isospin chemical potential looks qualitatively similar to the thermal dependence (left-hand panel). The increase in the isospin chemical potential is accompanied by a decrease in the normalized chiral quark condensates. This finding does not necessarily describe a linear and/or uniform dependence. In the hadronic phase, at $T = 100$ MeV, we find that $\sigma_u/\sigma_u^0$ and $\sigma_s/\sigma_s^0$ are almost not affected. Only at large $\mu_I$, the $\mu_I$-impacts are set . $\sigma_d/\sigma_d^0$ seems to be affected, earlier, namely, at lower values of $\mu_I$. It is noteworthy to highlight that even $\sigma_s/\sigma_s^0$ seems to respond to finite $\mu_I$. Its response, a decrease with an increase in $\mu_I$, emerges, at large $\mu_I$. We also find that the response of $\sigma_d/\sigma_d^0$ appears much earlier than that of $\sigma_u/\sigma_u^0$. From this phenomenological observation, we draw the conclusion that

- To each of the normalized chiral quark condensates, one could assign a *critical* isospin chemical potential so that

$$\mu_I^{(c)}\Big|_u < \mu_I^{(c)}\Big|_d < \mu_I^{(c)}\Big|_s, \tag{43}$$

and this is also valid in both the hadronic and partonic phases, as well as in the region of the phase transition, and

- The increasing temperature allows the phase transition related to the increasing isospin chemical potential to take place earlier and

$$T_\chi^{(c)}\Big|_u < T_\chi^{(c)}\Big|_d < T_\chi^{(c)}\Big|_s. \tag{44}$$

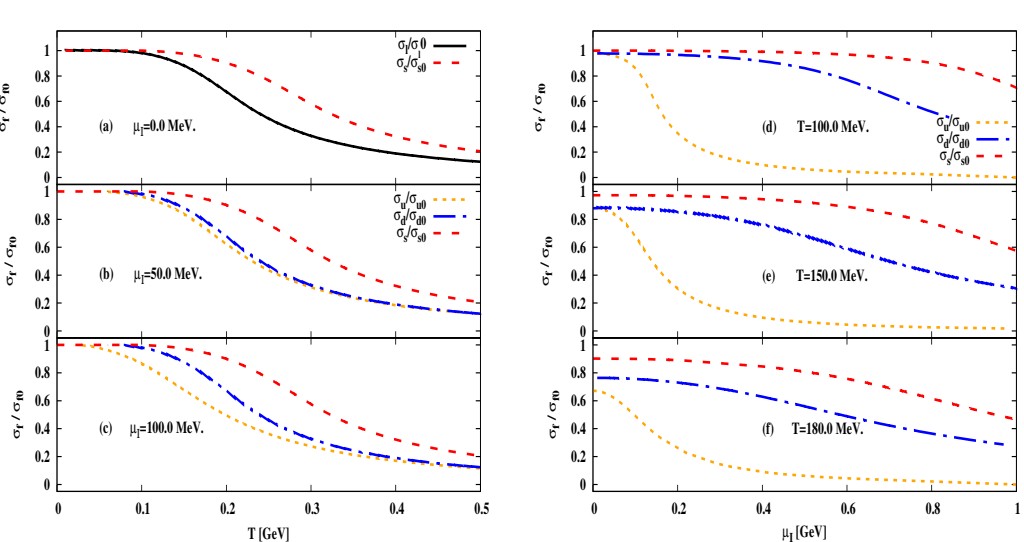

**Figure 1.** The dependence of the normalized chiral quark condensates on temperatures (left-hand panel) are compared with their dependence on the isospin chemical potentials (right-hand panel). The left-top panels show the results, at vanishing isospin chemical potential. The left-middle and left-bottom panels depict the results, at $mu_I = 50$ and 100 MeV, respectively. The right panels show the dependence on $mu_I$, at $T = 100$ MeV (right-top), $T = 150$ MeV (right-middle), and $T = 180$ MeV (right-bottom).

For the thermodynamic quantities, we start with the PLSM thermodynamic potential $\Omega(T, \mu_f)$, Equation (7). For example, the thermodynamic pressure is given as $p(T, \mu_f) = -\Omega(T, \mu_f)$. The left-hand panel of Figure 2 presents the PLSM calculations of the thermodynamic pressure as a function of the temperature $T$ normalized to $T_\chi$, the critical temperature, at vanishing isospin chemical potential. In this regard, we recall that at least two approaches are utilized in determining the chiral critical temperatures. The first one is the intersection point of the light quark and gluon condensates. The second one is the peak of the particle susceptibility. For more details, references [19,39,42] are recommended to interested readers.

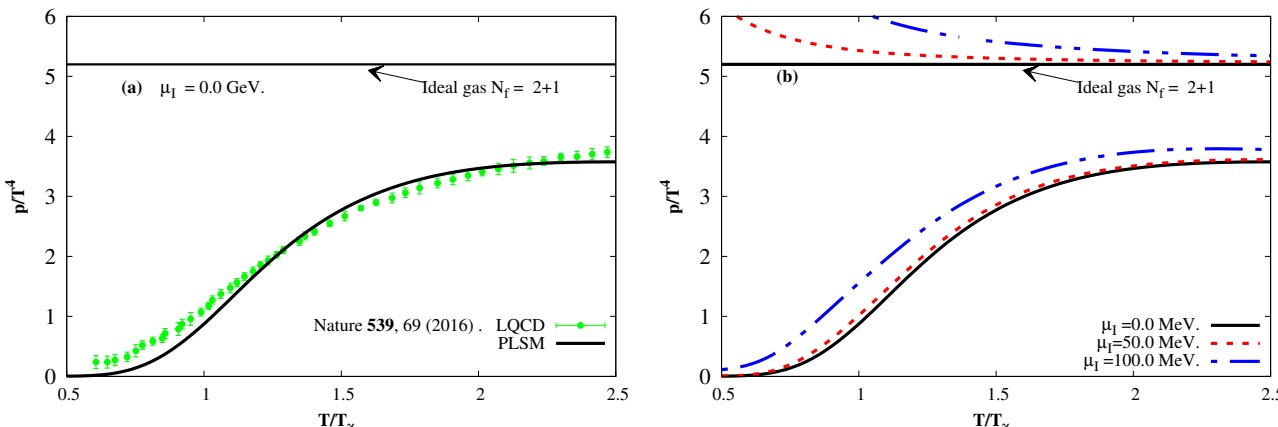

**Figure 2.** (**a**) Left-hand panel: the thermal dependence of the PLSM thermodynamic pressure compared with recent lattice simulations [55], at vanishing isospin chemical potential. (**b**) Right-hand panel compares the PLSM results at vanishing and finite isospin chemical potential.

The PLSM calculations at vanishing $\mu_I$ are compared with the recent lattice QCD simulations [55]. The right-hand panel depicts the same but at different isospin chemical potentials. We conclude that

- PLSM pressure agrees well with the lattice QCD results in both the hadronic and partonic phases,
- Both types of the phase transition in PLSM and lattice QCD are apparently identical and rapid crossover, and
- PLSM and lattice QCD seem to have comparable critical temperatures characterizing the hadron-quark phase transition.

In this regard, we recall that the PLSM critical temperature seems to be averaged from the various types of chiral quark condensations, $\sigma_f$, and the Polyakov-loop variables, $\Phi$ and $\bar{\Phi}$. The right-hand panel of Figure 2 compares the PLSM results at $\mu_I = 0$ MeV (slid curve), $\mu_I = 50$ MeV (dotted curve), and $\mu_I = 100$ MeV (dash-double-dotted curve). Although fixed temperatures are assumed, the increase in the isospin chemical potential seems to derive the colorless hadron system through crossover into the colored parton system.

### 3.2. In-Medium Modifications of $a_0$, $\sigma$, $\eta'$, $\pi$, $f_0$, $\kappa$, $\eta$, $K$, $\rho$, $\omega$, $\kappa^*$, $\phi$, $a_1$, $f_1$, $K^*$, and $f_1^*$ Meson States

The top panels of Figure 3 present the PLSM results on the thermal modifications of the scalar states $a_0$ (dashed curve) and $\sigma$ (dotted curve) and the pseudoscalar states $\eta'$ (solid curve) and $\pi$ (dashed-dotted curve). The middle panels depict the same but for the scalars $f_0$ (horizontal dashed curve) and $\kappa$ (vertical dashed curve) and the pseudoscalars $\eta$ (dotted curve) and $K$ (solid curve). The bottom panels present the thermal dependence of the vector mesons $\rho$ and $\omega$ (solid curve), $\kappa^*$ (long-dotted curve), and $\phi$ (dotted curve) and the axial-vector mesons $a_1 = f_1$ (dashed-dotted curve), $K^*$ (dotted curve), and $f_1^*$ (short dashed-dotted curve). The results at $\mu_B = 0$ MeV are depicted in the left-hand panel, while those at $\mu_B = 300$ MeV are depicted in the right panel . We observe no general tendency. However, in the partonic phase, i.e., at temperatures higher than the critical temperatures, we find that almost all masses increase with increasing temperature. The dense medium (right-hand panels) enhances the increase in the meson mass. We conclude that the temperature evolution of the various meson states, at high temperatures, forms a universal curve. In the hadronic phase, i.e., at temperatures below the critical values, we find that some meson masses remain constant, some decrease, and others seem to increase with the temperature. The dense medium largely contributes to this non-monotonic dependence. It is noteworthy to highlight that in the region of the phase transition, whose range of temperatures varies with the meson species, there is a prompt change in the meson masses.

Figure 4 presents the same mesons states as in Figure 3, $a_0$, $\sigma$, $\eta'$, $\pi$, $f_0$, $\kappa$, $\eta$, $K$, $\rho$, $\omega$, $\kappa^*$, $\phi$, $a_1$, $f_1$, $K^*$, and $f_1^*$, but here in dense medium, at fixed temperatures. The dependence on the baryon chemical potential is presented, at $T = 10$ MeV in the left-hand panel and at $T = 200$ MeV in the right-hand panel. Additionally, here we notice that the in-medium modifications of the meson masses vary with the meson species. At low densities, almost all meson masses slightly change with the increase in the baryon chemical potential. At large densities, there is a global tendency that the meson masses raise with the increase in the baryon chemical potential. We also observe that the density, i.e., the baryon chemical potential, seems to play almost the same role as that of the temperature, namely, deriving the colorless meson states through a region of phase transition, whose width varies with the meson species, into the colored partonic phase. Within the region of the phase transition, there is a prompt change in the meson masses. Of course, this phenomenon is especially apparent at low temperatures, i.e., in the bounded colorless meson states. At $T = 200$ MeV, the meson states are likely dissolved into the colored partonic phase, and therefore the increase in the masses is no longer depending on the meson species. We also find that the various mesons states seem to form a universal bundle, at large densities.

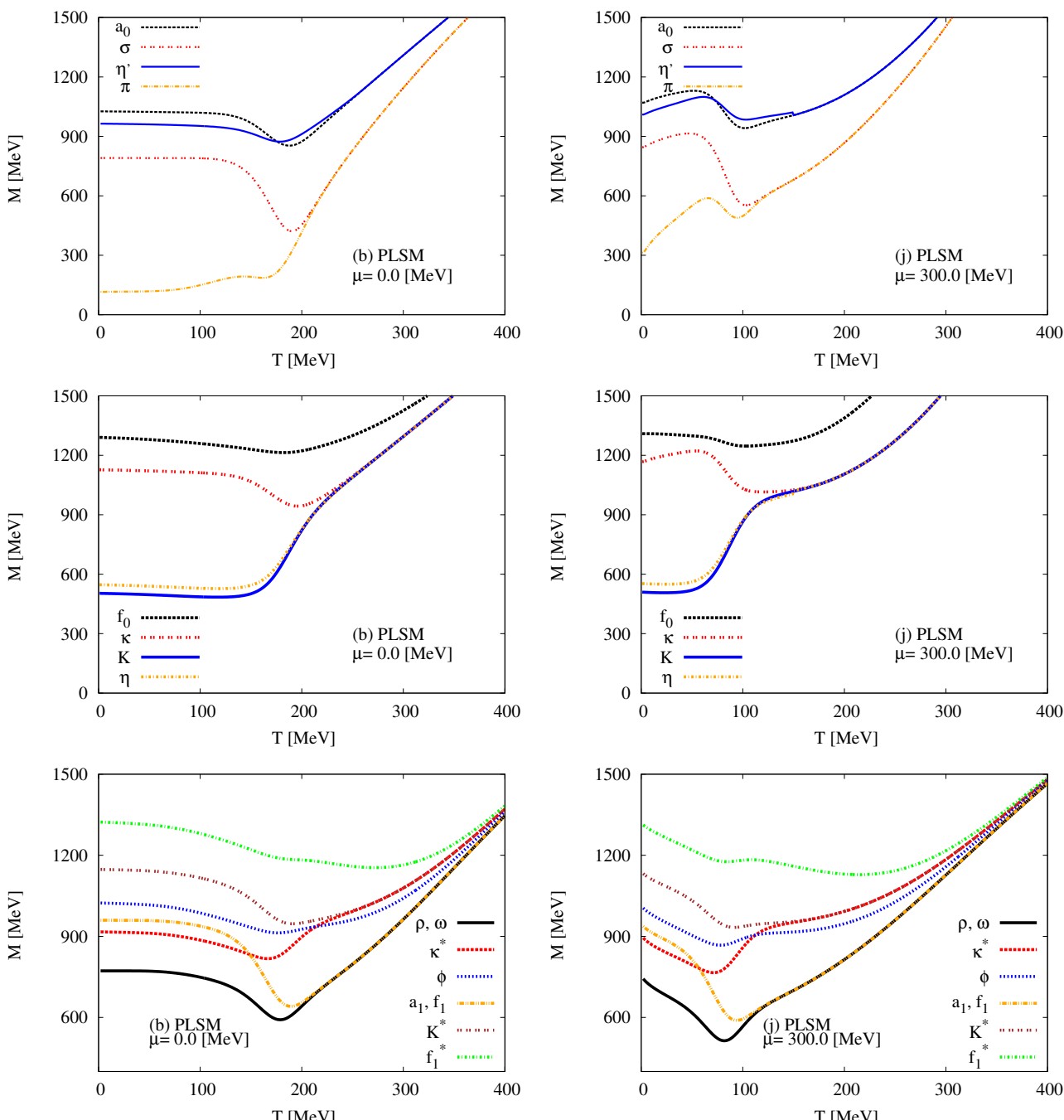

**Figure 3.** The in-medium medications of $a_0$, $\sigma$, $\eta'$, $\pi$ (top panels); $f_0$, $\kappa$, $\eta$, $K$ (middle panels); and $\rho$, $\omega$, $\kappa^*$, $\phi$, $a_1$, $f_1$, $K^*$, and $f_1^*$ meson masses. The left-hand panel depicts the in-medium thermal modifications of the meson masses, at $\mu_B = 0$ MeV. The left-hand panel presents the same but at $\mu_B = 300$ MeV.

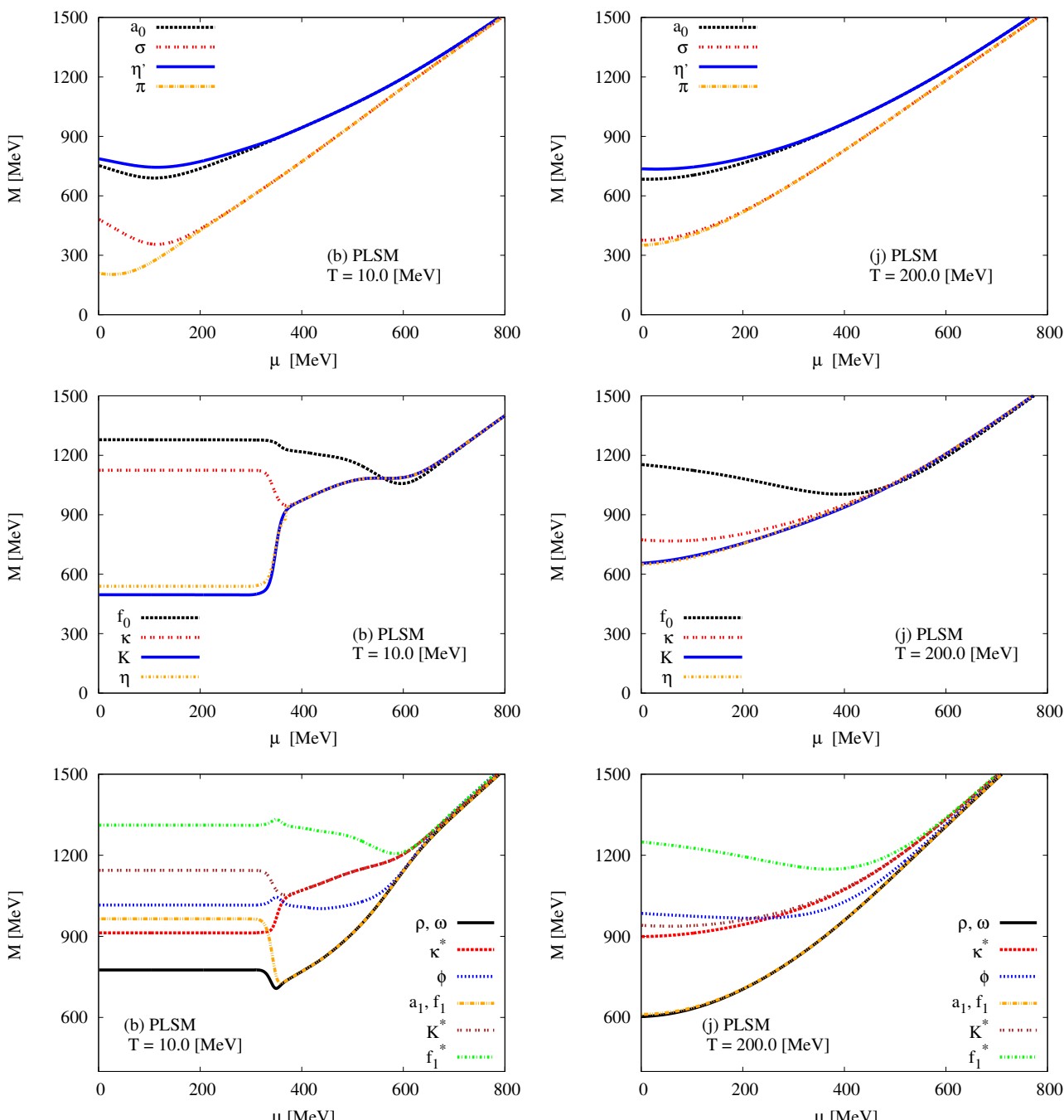

**Figure 4.** The same mesons states as in Figure 3, but here they are dependent on the baryon chemical potential, at $T = 10$ MeV (left-hand panel) and $T = 200$ MeV (right-hand panel).

### 3.3. QCD Phase Diagram at Finite Isospin Asymmetry

We find that the isospin asymmetry enhances the PLSM thermodynamic quantities, such as the pressure, especially at high temperatures. This observation can be analyzed by studying the dependence of the PLSM critical temperatures on the isospin chemical potential. Figure 5 depicts the $T - \mu_I$ phase diagram. The PLSM results are compared with the recent lattice QCD simulations [56–58]. It is apparent that the PLSM results agree well with the lattice QCD simulations. We also find that similar to $T - \mu_B$ phase diagram, the PLSM critical temperature decreases with the increase in the isospin chemical potential. For a reliable comparison, both temperature and isospin chemical potential are normalized to the corresponding critical temperature and pion mass, respectively.

For lattice QCD [56,57]: $m_\pi = 400.0$ MeV and $T_\chi^{\mu_I=0} = 164$ MeV. For PLSM: $m_\pi = 138$ MeV and $T_\chi^{\mu_I=0} = 210$ MeV. When comparing Figure 5 with the QCD phase structure in $(T - \mu_B)$-plane reported in references [3,31,39,41], an excellent similarity, at least qualitative, could be concluded.

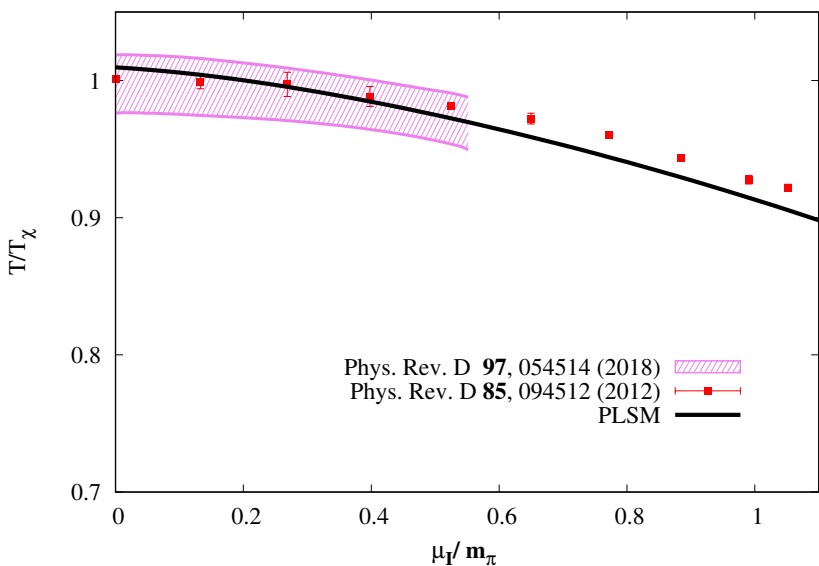

**Figure 5.** The QCD phase diagram of critical temperature and finite isospin chemical potential, at vanishing baryon chemical potential. The PLSM results (solid curves) are compared with the recent lattice QCD calculations (symbols) [56,57].

## 4. Conclusions

In studying QCD phase structure, thermodynamics and in-medium modifications of the masses of $a_0, \sigma, \eta', \pi, f_0, \kappa, \eta, K, \rho, \omega, \kappa^*, \phi, a_1, f_1, K^*$, and $f_1^*$ meson states, we utilized the Ployakov linear-sigma model with three quark flavors. The medium is QCD matter characterized by finite temperatures, baryon chemical potentials, and isospin chemical potentials. The finite isospin asymmetry causes the emergence of nonvanishing diagonal generators, namely, $\sigma_0 \neq \sigma_3 \neq \sigma_8 \neq 0$ of the mean sigma-fields $\bar{\sigma}_a$, where $a \in \{0, 1, \cdots, 8\}$. This means that the SU(2) isospin asymmetry is broken through $\sigma_3$, i.e., $\sigma_u = \sigma_l + \sigma_3$ and $\sigma_d = \sigma_l - \sigma_3$ [50,59,60]. The inclusion of interactions in the PLSM approach causes the emergence of the U(1)$_A$ anomaly. From the thermal and dense dependence of the chiral quark condensates $\sigma_u$, $\sigma_d$, and $\sigma_s$, the QCD phase structure could be characterized. These quantities are to be estimated by minimizing the thermodynamic potential, $\Omega(T, \mu_f)$, Equation (7). The excellent agreement between the present PLSM thermodynamics and recent lattice QCD simulations, at finite isospin chemical potential, promotes the conclusion that PLSM helps in obtaining a clear picture of the finite baryon-density QCD, at least qualitatively. We conclude that the dependence of the critical temperatures on the isospin chemical potentials, the phase diagram of finite-isospin-QCD, looks very similar to the dependence of the critical temperatures on the baryon chemical potentials, the phase diagram of finite-baryon-dense-QCD.

From the QCD thermodynamics, we conclude that PLSM thermodynamic quantities, for example, the pressure, agree with the lattice QCD simulations over the entire range of temperature combining both the hadronic and partonic phases. We also conclude that the type of the phase transition in the PLSM and lattice QCD simulations is similar, rapid crossover . In this regard, we find that PLSM and lattice QCD have comparable critical temperatures characterizing the hadron-quark phase transition.

From the in-medium modifications of the masses of $a_0, \sigma, \eta', \pi, f_0, \kappa, \eta, K, \rho, \omega, \kappa^*, \phi, a_1, f_1, K^*$, and $f_1^*$ meson states, we conclude that the thermal and dense dependence of the meson masses is nonmonotonic. We find that, at large temperatures and chemical

potentials, the confined meson states likely dissolve into deconfined parton matter so that the various meson masses form a universal bundle combining most of the meson states. We also conclude that the increase in the chemical potential derives the colorless confined meson states into colored deconfined parton matter. Within the temperatures or chemical potentials range of phase transition, the change in the meson masses is large.

**Funding:** This research received no external funding

**Data Availability Statement:** The authors confirm that the data supporting the findings of this study are available within the article.

**Conflicts of Interest:** The authors declare no conflict of interest

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
