# Peer review of "QCD Phase Structure and In-Medium Modifications of Meson Masses in Polyakov Linear-Sigma Model with Finite Isospin Asymmetryâ€"

_universe, doi:10.3390/universe9060276_

Round 1
Reviewer 2 Report
The author discusses the QCD phase diagram in the (T − μ_I) plane and finds that this diagram is qualitatively similar to the (T − μB) phase diagram. The study is done within the scope of the Polyakov-loop effective model in which the couplings are fixed on phenomenological grounds. Meson masses are calculated.
The paper is well-written, and the methods are adequately described. The results matched well with first-principle lattice calculations.
I recommend the publication of this paper. I suggest that the author should address minor English issues (indicated in separate comments) and also improve the style of equations (by rearranging them) at lines 71, 95, 118, 125, 136, 147, 159, and 165.
The English of the paper is very good. I would draw the attention of the Author to the following lines and phrases, to be reformulated:
- 130 are related to the degrees of 130 freedom, the quark flavors, integrated in.
- 153: the meson potential constructing the mass matrix.
- 231: We conclude that the various meson states 231 form a universal bundle, at large temperatures.
(please, explain the meaning of "bundle").
Reviewer 3 Report
The subject is important for hadron physics, and the paper should be published. But I regret the paper is not well written. I guess that the author did not have enough time to improve the paper.
1. Abstract, 3rd line. natural --> neutral?
2. Keywords. too general to specify the content. For example, "Chiral Lagrangian".
3. Fifth line from the bottom of section1. axial-cector --> axial-vector ?
4. Equation number (9) is not displayed. Better to break the line for example before 4th term (lambda_2 term)
5. Equation number (17) is not displayed. Better to break between 2nd and 3rd terms. (f_K^+- and f_K^0)
6. I cannot find the equation numbers from (24) to (28).
7. The next equation of (34) ((35)?), the right hand of the equation exceeds the page.
8. Same for the right-hand of m^2_i,ab
9. Figure caption of Fig.1, (left-hand panel) (right-hand pane)
Are they "upper panel" and "lower panel"?
Consequently, "(right-top)" and "(right-bottom)" should be changed.
10. They are 48 references. From 1. to 4, 6 and 7, 18 to 30, they are the authors papers.
The author can select the references, or he should give brief explanation for each.
Otherwise, the readers think the references are put because the author want to increase his citation.
It is really sad that most readers stop reading this paper halfway through reading, although the paper is very valuable.
enough readable as a scientific paper.
Round 2
Reviewer 1 Report
see attached file

Reviewer 3 Report
The paper studies an important subject, and now, the paper is more readable. I wish the author to continue the research.